# A Potential ABA Analog to Increase Drought Tolerance in *Arabidopsis thaliana*

**DOI:** 10.3390/ijms24108783

**Published:** 2023-05-15

**Authors:** Ruiqi Liu, Guoyan Liang, Jiaxin Gong, Jiali Wang, Yanjie Zhang, Zhiqiang Hao, Guanglin Li

**Affiliations:** 1Key Laboratory of Ministry of Education for Medicinal Plant Resource and Natural Pharmaceutical Chemistry, College of Life Sciences, Shaanxi Normal University, Xi’an 710119, China; 2National Engineering Laboratory for Resource Development of Endangered Crude Drugs in Northwest China, College of Life Sciences, Shaanxi Normal University, Xi’an 710119, China

**Keywords:** abscisic acid, virtual screening, PYR1/PYLs, drought stress

## Abstract

Abscisic acid (ABA) plays an important role in the response of plants to drought stress. However, the chemical structure of ABA is unstable, which severely limits its application in agricultural production. Here, we report the identification of a small molecule compound of tetrazolium as an ABA analog (named SLG1) through virtual screening. SLG1 inhibits the seedling growth and promotes drought resistance of *Arabidopsis thaliana* with higher stability. Yeast two-hybrid and PP2C inhibition assays show that SLG1 acts as a potent activator of multiple ABA receptors in *A. thaliana*. Results of molecular docking and molecular dynamics show that SLG1 mainly binds to PYL2 and PYL3 through its tetrazolium group and the combination is stable. Together, these results demonstrate that SLG1, as an ABA analogue, protects *A. thaliana* from drought stress. Moreover, the newly identified tetrazolium group of SLG1 that binds to ABA receptors can be used as a new option for structural modification of ABA analogs.

## 1. Introduction

Drought is one of the most severe abiotic stresses that greatly restricts plant growth and crop production [1]. Abscisic acid (ABA) is a crucial phytohormone that functions during the adaptive response to drought; it also plays an essential role in seed germination, dormancy, and plant growth [2,3]. ABA enhances plant tolerance to drought by regulating stomatal aperture, plant transpiration, and water absorption by roots [2,4]. The rational use of abscisic acid is of great significance for improving the drought resistance of crops.

In *Arabidopsis thaliana*, researchers have now identified the core ABA signaling components, including ABA receptor (RCAR)/pyrabactin resistance (PYR1)/PYR1-like proteins (PYLs), clade A type 2C protein phosphatases (PP2Cs), sucrose non-fermenting-1 (SNF1)-related protein kinase 2s (SnRK2s), the ABA-responsive element binding factors (ABFs), and the downstream proteins [5,6]. The PYR1/PYLs contain a ligand-binding pocket flanked by two highly conserved loops, named as gate/CL2/β3-β4 loop and latch/CL3/β5-β6 loop [6,7]. Upon binding by ABA, PYR1/PYLs undergo conformational rearrangements and form complexes with PP2Cs to relieve the inhibitory effects of PP2Cs on proteins SnRK2s, thereby activating ABA signaling [5,8]. In *A. thaliana*, 14 PYL family members are distributed into three subfamilies, Subfamily I (PYL7/8/9/10), II (PYL4/5/6/11/12/13), and III (PYR1, PYL1/2/3), of which PYL13 is ABA irresponsive [9,10]. Despite the functional redundancy of PYL members in Arabidopsis, they also exhibit unique characteristics in the ABA sensitivity and oligomerization status. Subfamily III receptors PYR1 and PYL1/2/3 are essential ABA-dependent inhibitors of PP2Cs, among which PYR1, PYL1, and PYL2 are constitutive dimers, whereas PYL3 may be in a state of equilibrium between dimers and monomers [11]. Subfamily I (PYL7/8/9/10) and II (PYL4/5/6/11/12) receptors are monomers that inhibit the phosphatase activities of PP2Cs even without the participation of ABA. However, they are still regulated by ABA because these PYLs can completely inhibit activities of PP2Cs at much lower concentrations in the presence of ABA [11]. 

Due to the vital role of ABA in drought response, various studies have proven that exogenous application of ABA improves plant drought tolerance. However, ABA is unstable and decomposes rapidly in plants, severely limiting its agriculture applications [12]. Therefore, the development of ABA analogs may have important agricultural applications [13]. Multiple methods have been used to screen ABA analogs from small-molecule libraries. For instance, quinabactin (QB), also known as ABA mimic 1 (AM1), was identified by yeast two-hybrid assays and AlphaScreen assays from 57,000 and 12,000 compounds, respectively [14,15]. QB is a sulfonamide ABA agonist of mainly dimeric PYLs, which induced the closure of the guard cells of *A. thaliana* and soybean leaves, inhibited water loss, and improved plant drought tolerance [14,15]. An ABA functional analog SIMILAR TO ABA 7 (S7) was identified by using the dual-luciferase system driven by the promoter of rice ABA-responsive genes from 55,000 small-molecule compounds [16]. However, the experimental identification of ABA analogues usually requires the use of expensive chemical libraries and high-throughput screening equipment. 

With the emergence of computer-aided drug design tools, the discovery of target compounds has become efficient and cheap through virtual screening technologies, including molecular docking and molecular dynamics based on the three-dimensional structure of proteins [17]. Elucidation of PYLs crystal structures provides a basis for use of virtual screening [7]. Vaidya et al. used virtual screening technologies combined with X-ray crystallography and structural designing to develop an amide compound opabactin, whose activity is ten-fold higher than ABA in both monocotyledonous and dicotyledonous plants from 18 million compounds [18]. Recently, through structural targeted design and structural modification, a sulfonamide sulfobactin (SB) derivative iSB09 was identified as an ABA agonist for the engineered CsPYL1 ABA receptor, named CsPYL15m CsPYL1^5m^ [19]. Therefore, the use of virtual screening technology to develop new ABA analogues is of great significance for improving the drought tolerance of crops.

In our study, we used molecular docking, pharmacophore screening, experimentation, and molecular dynamics to identify and characterize a small molecule compound of tetrazolium as an ABA analog (named SLG1) from about 1 million compounds. SLG1 was found to inhibit the growth of seedlings and enhance drought resistance in *A. thaliana*. Unlike the previously screened ABA analogs that bind to PYR1/PYLs through their sulfamide or amide structure, SLG1 binds to PYL1 through a tetrazole group, which enriches the types of ABA analogues and provides some new ideas for the subsequent structural modification of ABA analogues. These results show that the combination of virtual screening and experimentation is an effective method for screening ABA analogues to protect plants under drought stress.

## 2. Results

### 2.1. Ten Small Molecule Compounds Are Screened as Potential ABA Analogues through Virtual Screening Techniques

To identify ABA analogues using virtual screening techniques, we first downloaded the co-crystal structures of pyrabactin-PYL1 (PDB_ID: 3NEF) from the PDB library and re-docked ABA to the PYL1 to test the reliability of the active site (Figure 1A–C). Then, we performed receptors-based virtual screening using LibDock and CDOCKER modules of the Discovery Studio software (DS). As a result, about 13,000 small-molecule compounds with a score greater than 120 were screened from about 1 million compounds using LibDock modules. Next, 353 small-molecule compounds with -CDOCKER Interaction energy greater than 50 and -CDOCKER energy greater than 38.3461 were screened using CDOCKER modules from 13,000 compounds obtained in the previous step. 

In addition, we performed further screening of 353 small molecule compounds using a ligand-dependent pharmacophore modeling. Ten pharmacophores were constructed based on 55 feature elements extracted from S-(+)-ABA, pyrabactin, and quinabactin (QB), including 29 hydrogen bond acceptors, 4 hydrogen bond donors, 11 hydrophobic centers, 1 positive charge center, and 10 aromatic ring centers (Figure 1D; Appendix A). The optimal pharmacophore was determined by matching compounds in the test set to ten pharmacophore models. The results showed that pharmacophore 07 had a high degree of matching with the compounds in the test set, and also had a good match with ABA (Figure 1E,F). Therefore, pharmacophore 07 was used to screen potential ABA analogs from 353 small-molecule compounds. Finally, ten small-molecule compounds were screened out as potential ABA analogues, named SLG1–10 (Appendix A).

### 2.2. SLG1 Inhibits the Seedlings Growth of A. thaliana and Activates the Expression of ABA-Related Genes

To test whether SLG1-10 performs physiological functions similar to ABA, we investigated the in vivo activities of SLG1-10 on *A. thaliana* wild-type Col-0 (WT) by directly placing the seeds in half strength of MS media containing different concentrations of compounds. As a result, we found that one of the 10 compounds, SLG1, a tetrazolium-like compound, inhibited the seedling growth of *A. thaliana* seedlings (Figure 2A,B). The growth of *A. thaliana* seedlings was almost completely inhibited at 1 μM concentration of ABA. The growth of *A. thaliana* seedlings was also significantly inhibited by SLG1 in a concentration dependent manner, as evidenced by the fact that the length of primary roots decreased by about half at 100 μM and 200 μM concentration of SLG1 compared to the control (DMSO) (Figure 2A,B). Then, the expression levels of four ABA-responsive genes *ABF2* [20], *RD29b* [21], *KIN1* [22], and *P5CS1* [23] were analyzed by quantitative real-time PCR (qRT-PCR) in the above-mentioned seedlings. qRT-PCR analysis revealed that the four genes were induced significantly at 1 μM of ABA (Figure 2C–F). Meanwhile, at low concentrations of SLG1 (1–50 μM), two genes *ABF2* and *P5CS1* were significantly induced and at high concentrations of SLG1 (100 and 200 μM), four genes *ABF2*, *RD29b*, *KIN1*, and *P5CS1* were significantly activated (Figure 2C–F). These results suggested that SLG1may be a potential ABA functional analogue.

### 2.3. SLG1 Reduces Water Loss and Enhances Drought Resistance

ABA is reported to enhance drought tolerance by inhibiting the stomatal opening and reducing transpiration in plants [2]. Therefore, the stomatal aperture, water loss, and drought tolerance in *A. thaliana* were analyzed after treatment with different concentrations of SLG1, using ABA and DMSO as the control. As shown in Figure 3A, 10 and 25 μM of ABA significantly decreased the stomatal aperture of leaves in *A. thaliana* compared to DMSO treatment. Similarly, the stomatal aperture of *A. thaliana* was also significantly reduced after 10–200 μM of SLG1 treatment and the effect of SLG1 on stomatal movement showed a concentration-dependent manner (Figure 3A). Moreover, water loss from the leaves also was significantly decreased after SLG1 (50, 100, and 200 μM) and ABA (50 μM) treatments compared with DMSO, although the effect of SLG1 was less than that of ABA (Figure 3B). Consistent with water loss, 50–200 μM of SLG1 and 50 μM of ABA dramatically enhanced drought tolerance of *A. thaliana* seedlings, and the survival rate after drought stress was greater for SLG1-treated and ABA-treated plants than for the DMSO-treated plants (Figure 3C,D). These results suggested that SLG1 increases the drought tolerance of *A. thaliana*. 

Generally, ABA is unstable and decomposes rapidly in plants, limiting its application in agriculture [12]. SLG1 comprises two chemical groups: ester and tetrazolium, different from ABA’s carboxylic acid and ketone groups (Figure 1D; Appendix A). To test the stability of SLG1 and ABA, 100 μM of SLG1 and ABA were exposed to UV light (8 W) and their levels were quantified using LC-MS. With the increase in UV exposure time, the concentration of ABA dropped rapidly during the initial 4–16 h of irradiation and was close to zero at 48 h, while the concentration of SLG1 remained unchanged (Figure 3E). These results indicated that SLG1 is more stable than ABA.

### 2.4. SLG1 Is a Potential agonist of Multiple ABA Receptors

The ABA receptor family consists of 14 START domain proteins in *A. thaliana*, including ABA-responsive PYR1/PYL1–12 and ABA-irresponsive PYL13 [10,24]. To investigate the activity of SLG1 to the ABA receptor family, we analyzed the interaction between PYR1/PYL1–12 receptors and three clade A PP2Cs (ABI1, ABI2, and HAB1) after SLG1 treatment by yeast two-hybrid assays. At 10 μM, ABA promoted the interaction of multiple ABA receptors, especially ABA-dependent PYR1/PYL1/PYL2/PYL3 with ABI2 (Figure 4A), PYR1/PYL1/PYL2/PYL4 with ABI1 (Appendix A), and PYL1/PYL2/PYL11 with HAB1 (Appendix A). Similarly, 10 μM of SLG1 significantly promoted the interaction of PYL2/PYL3 with ABI2 (Figure 4A), PYL2 with ABI1 (Appendix A), and PYL2/PYL11 with HAB1 (Appendix A). As the concentration of SLG1 increased (50, 100, 150, and 200 μM), no significant change was observed in the interaction between PYLs and PP2C (Figure 4A; Appendix A). Subsequently, PP2C inhibition assays were performed to further analyze their interaction (Figure 4B). After treatment with 10 µM of ABA, the ABI2 PP2C activity was significantly inhibited by PYR1/PYL1/PYL2/PYL3, moderately suppressed by PYL6/PYL7/PYL9, and slightly inhibited by PYL4/PYL5/PYL8/PYL10/PYL11/PYL12 (Figure 4B). After treatment with 10 µM of SLG1, PYL2/PYL3/PYL7 significantly inhibited the ABI2 PP2C activity, while PYL6/PYL9 moderately suppressed the activity and PYL12 slightly inhibited the activity (Figure 4B). Yeast two-hybrid assays and PP2C inhibition assays showed that SLG1 acts as a potential activator of multiple ABA receptors, especially PYL2 and PYL3 in *A. thaliana*.

### 2.5. Structural Basis of SLG1 Recognition by ABA Receptors

To understand the structural basis of SLG1 as an ABA receptors’ agonist, we determined and compared the two-dimensional and three-dimensional structure of SLG1-PYL2/PYL3 and ABA-PYL2/PYL3 complexes. ABA was mainly bound to PYL receptors through its carboxyl group (Figure 5A,C,E,G), while SLG1 was bound to PYL2 and PYL3 mainly through its tetrazolium group (Figure 5B,D,F,H). The residue K^64^ of PYL2 forms a salt bridge and two hydrogen bond interactions with N of the SLG1 etrazolium group, while K^64^ of PYL2 forms a salt bridge and one hydrogen bond interaction with O of ABA carboxyl group (Figure 5A–D). ABA forms hydrophobic interactions with multiple residues (Phe^66^, Val^87^, Ala^93^, Val^166^, His^119^, Leu^121^, Tyr^124^, and Val^169^) of PYL2, but SLG1 forms less hydrophobic interaction with Lys^64^, Val^67^, and Val^85^ residues of PYL2 (Figure 5A–D). Similarly, the tetrazolium group of SLG1 mainly forms two salt bridges with K^79^ of PYL3 and one hydrogen bond interaction with Arg^103^ of PYL3, two ester groups of SLG1 forms two hydrogen bond interactions with Arg^103^ and Asn^196^, while the carboxyl group of ABA forms a salt bridge and one hydrogen bond interaction with K^79^ of PYL3 (Figure 5E–H). Moreover, ABA also forms hydrophobic interactions with multiple amino acid residues (Leu^111^, Val^107^, Val^134^, Tyr^144^, Val^189^, Val^192^, and Val^193^) of PYL3, while SLG1 does not form hydrophobic interactions with PYL3 (Figure 5E–H). In addition, the binding conformation of SLG1 in the receptors PYL2 and PYL3 is highly similar, of which its tetrazolium group is all located on the innermost side of the receptor active site (Figure 5B,D,F,H). The binding of SLG1 to PYL2/PYL3 is weaker than that of ABA, and the binding of SLG1 to PYL2 is stronger than that of PYL3 (Figure 5). These results indicated that the tetrazolium group of SLG1 mimics the carboxyl group of the ABA to combine PYLs, which provides a structural explanation for their shared biological roles in plant stress responses and the weaker role of SLG1 than ABA.

### 2.6. Molecular Dynamics Simulations of SLG1 Recognition by ABA Receptors

MD simulations (50 ns) were executed to evaluate the binding and stability attributes of ABA-PYL2/PYL3 and SLG1-PYL2/PYL3 complexes [25]. First, the root mean square deviation (RMSD) of the backbone Cα atoms of the complex was used to check the conformational stability of the ligand–protein complexes. Results indicated that RMSD values of SLG1-PYL2, SLG1-PYL3, ABA-PYL2, and ABA-PYL3 complexes fluctuates slightly and remain stable throughout the simulation period compared to the individual protein Apo (Reference) (Figure 6A,B). Furthermore, the root mean square fluctuation (RMSF) was measured to assess the flexibility of ligand–protein binding. When SLG1 and ABA bind to PYL2, RMSD values of about 1–70 and 163–170 residues of PYL2 are higher than the reference, and RMSD values of the other residues of PYL2 are approximately the same as the reference (Figure 6C). When SLG1 and ABA bind to PYL3, RMSF values of almost all amino acid residues in PYL3 are consistent with that of the reference (Figure 6D). Subsequently, the radius of gyration (Rg) was used to evaluate the compactness of the complex. For SLG1-/ABA-PYL2 combinations, the Rg fluctuations are less than 0.1 nm throughout the 50 ns and basically the same as the reference (Figure 6E). For SLG1-/ABA-PYL3, the fluctuations of Rg values are also less than 0.1 nm throughout the 50 ns and their fluctuations are smaller than the reference (Figure 6F). Finally, the solvent-accessible surface area (SASA) was used to evaluate the stability of protein folding. SASA values of ABA-PYL2 and SLG1-PYL2 complexes fluctuate little throughout the entire 50 ns with an average of 107 nm^2^, which is slightly higher than the reference (Figure 6G). For ABA-PYL3 and SLG1-PYL3 complexes, their SASA values fluctuate very little and are consistent with the reference (Figure 6H). These results indicated the ABA-/SLG1-PYL2/PYL3 complexes are stable throughout the simulation process. 

## 3. Discussion

Drought stress is a major threat to crop production, but effective methods to alleviate the adverse effects of drought are not available. The plant hormone ABA plays central roles in adaptive responses to abiotic stresses, particularly drought [3]. Due to the fact that ABA is unstable and decomposes rapidly in plants, ABA mimics are being investigated as tools for alleviating the impact of drought stress on crop yields [13]. Chemical biology and structure-based virtual screening approaches have enabled the discovery of an important number of ABA analogues, e.g., pyrabactin [26], AM1 [14] or QB [15], AM1 fluorine derivatives AMF4 [27], and opabactin [18]. In this study, we used both receptor-based and ligand-based virtual screening approaches to screen for ABA analogues. LibDock, a receptor-based rigid molecule docking tool, can rapidly screen target compounds from the compound database, and CDOCKER, a semi-flexible molecular docking tool based on CHARMm’s position, can produce highly accurate docking results [28]. As a result, 353 candidate target compounds were quickly and accurately screened from approximately 1 million small molecule compounds using both LibDock and CDOCKER. In addition to using receptor-based molecular docking, ligand-based pharmacophore modeling was also established based on the common chemical characteristics of a group of ligands (ABA, pyrabactin, and QB) [29]. Based on this, we screened 10 compounds from 353 candidate target compounds for subsequent experimental verification and finally identified SLG1 as an ABA analogue (Figure 7). Our studies provide an effective method to screen ABA analogues from the database. 

Our results showed that SLG1 can inhibit the growth of *A. thaliana* seedlings in a concentration-dependent manner (Figure 2A,B). Additionally, SLG1 can reduce the water loss of separated leaves and stomatal conductance (Figure 3A,B). Moreover, SLG1 can protect *A. thaliana* from drought stress (Figure 3C,D). These results suggested that SLG1 is a potential ABA analogue, but its low activity is relative to ABA. This may be because PYL selectivity of SLG1 is different from that of ABA, which is a pan agonist of the PYR1/PYLs family of receptors [30]. In this study, a yeast two-hybrid experiment showed that SLG1 can promote the interaction of PYL2-ABI1/ABI2/HAB1, PYL3-ABI2, and PYL11-HAB1, while ABA can promote the interaction of PYR1-ABI1/ABI2, PYL1-ABI1/ABI2/HAB1, PYL2-ABI1/ABI2/HAB1, and PYL3-ABI2 (Figure 4A and Appendix A). PP2C inhibition assays of ABI2 also showed that ABA is an agonist of most ABA receptors, while SLG1 is mainly an agonist of PYL2/PYL3/PYL7 (Figure 4B). Other ABA analogues also have PYL selectivity. For example, AM1 specifically promotes the interaction of PYR1, PYL1-3, PYL5, and PYL7 with HAB1, which explains why AM1 is weaker than ABA in activating ABA related gene expression and enhancing plant drought resistance [14]. 

At present, multiple ABA analogs, such as AM1 [14], AMF4 [27], julolidine- and fluorine-containing ABA receptor activator (JFA1 and JFA2) [31], opabactin [18], and iSB09 [19], have a sulfonamide group, and their sulfonamide group resembles the carboxylate group of ABA. Combining yeast two-hybrid assays and PP2C inhibition experiments, it was found that SLG1 is mainly an agonist of PYL2 and PYL3. Therefore, molecular docking of SLG1 bound to PYL2/PYL3 is performed. After docking SLG1 and PYL2/PYL3, we found that the combination of SLG1 and PYL2/PYL3 is neither dependent on the carboxyl group nor the sulfonamide group, but on the tetrazolium group (Figure 5). As a hydrogen bond acceptor, the tetrazolium group forms a strong hydrogen bond interaction with the active site of PYL2/PYL3 in the deepest part of the pocket. Thus, the tetrazolium group can be used as a new option for structural modification of ABA analogs. 

Because structural instability of ABA severely limits its application in agricultural production, therefore, stability of the compound SLG1 was measured. The metabolic instability of ABA is due to glycosylation at its carboxyl group and hydroxylation at its cyclohexenone ring [32,33]. The light sensitivity of ABA is due to the conjugated linker between its carboxylate and cyclohexenone ring [14]. SLG1 is composed of the tetrazolium ring, not containing the carboxyl group and cyclohexenone (Figure 7). In this study, ABA degrades slightly when exposed to UV light for 1 h, rapidly degrades at 4 h, and basically degrades to below 10% at 48 h, while the structure of SLG1 is still stable when exposed to UV light for 48 h (Figure 4E), supporting that SLG1 is chemically more stable than ABA. 

In conclusion, we identify SLG1 as a stable ABA analogue that can protect plants from water loss and enhance drought tolerance. Additionally, SLG1 efficiently binds to multiple ABA receptors via its tetrazolium group, thus providing a new option for structural modification of ABA analogs. Given the highly conserved nature of ABA receptors and their signaling mechanism in crop plants, SLG1 may be used to improve drought tolerance in economically important crops. 

## 4. Materials and Methods

The schematic flow chart describing each step of the research methodology is illustrated in Figure 7.

### 4.1. Preparation Crystal Structure of PYL1 Protein

The structural composition of PYL1 with the agonist pyrabactin (PDB_ID: 3NEF) was downloaded from the Protein Data Bank (http://www.pdb.org, accessed on 1 January 2019). PYL1 is a dimeric receptor composed of the A and B chains: The A chain is bound to the ligand. The B chain was deleted, and the A chain was used for molecular docking. The crystal structure of PYL1 was optimized using the Prepare Protein module of the Discovery Studio software (v2017), including deleting excess water molecules, adding residues to incomplete amino acids, adding charges by the CHARMm force field, and specifying the proton status of the residues. In addition, the active site for molecular docking was determined according to the position where the ligand pyrabactin bound to PYL1, and the original ligand was deleted. The size of the active pocket was set to 20 Å to allow efficient binding of the ligand to the PYL1 protein. 

### 4.2. Preparation of Small Molecule Compounds

About 980,000 compounds were downloaded from the small molecule compound library of Maybridge (https://www.maybridge.com, accessed on 1 January 2019) and Life Chemical (https://lifechemicals.com, accessed on 1 January 2019). The structures of these compounds were first optimized using the Prepare Ligand module of Discovery Studio software by adding charges with the CHARMm force field, generating tautomers and converting them into 3D structures. This optimized structure was energy-minimized for virtual screening.

### 4.3. Molecular Docking Screening

Molecular docking of the LibDock module was carried out using the following parameters: ‘Docking Preferences’ was set to ‘fast search’ and other parameters are default. The final ranking of small-molecule compounds was based on the LibDock Score from the docking. Small-molecule compounds with the LibDock Score greater than 120 were screened for further screening using the semi-flexible molecular docking CDOCKER module of the DS software [28]. The parameter settings are as follows: ‘Pose Cluster Radius’ was set to 0.5 and other parameters are default. Small-molecule compounds with -CDOCKER Interaction energy greater than 50 and -CDOCKER energy greater than 38.3461 were screened for further study.

### 4.4. Pharmacophore Screening

The structure of pyrabactin, QB, and (+)-abscisic acid was downloaded from the ZINC library and used as the training set. These molecules were docked to the PYL1 receptor to search for their optimal conformation. The Principal and MaxOmitFeat values of the three compounds were 2 and 0, respectively. Then, the pharmacophore model was constructed using the Common Feature Pharmacophore module of the DS software. The model contained six types of feature elements, including hydrogen bond acceptor (HB_ACCEPTOR), hydrogen bond donor (HB_DONOR), hydrophobic center (HYDROPHOBIC), positive charge center (POS_IONIZABLE), negative charge center (NEG_IONIZABLE), and aromatic ring center (RING_ARMATIC). The pharmacophore model was constructed according to the following parameters: the upper limit of the best conformation was set as 200 and the energy threshold was set as 10, resulting in a total of ten pharmacophore models. 

Ten small-molecule compounds reported to bind to PYR1/PYLs were downloaded from the ZINC library. They were used as the test set to screen the best pharmacophore models using the Ligand Profiler module with default parameters. Finally, the Screen Library module of DS software was used to screen the compounds obtained in the second round. The match between the small molecules and the best pharmacophore was judged: the higher the FitValue, the better the match. 

### 4.5. Phenotypic Assays

ABA and SLG1-SLG10 with a purity greater than 98% were purchased from TOPSCIENCE. The growth of seedlings was carried out as described by Li et al. [34]. Sterilized *A. thaliana* Col-0 seeds were stratified for 3 days at 4 °C on 1/2 MS medium supplemented with 1 μM ABA or various concentrations of SLG1, using 0.5% DMSO as the control, and then grown for another seven days in a long-day light incubator (16 h light, 8 h dark). The root length of seedlings was measured with Image J.

Stomatal conductance was measured using the previously reported method [35]. Seven-day-old wild-type Arabidopsis with identical sizes were transferred to soil and allowed to grow under long-day conditions for another two weeks. The leaves were cut out carefully, soaked in MES-KOH solution (10 mM MES-KOH, 10 mM KCl, 0.05 mM CaCl_2_, pH 6.15), and placed in a light incubator for 3 h to fully open the stomata. Then, SLG1 at different concentrations was added to the above solution, using DMSO as a negative control and ABA (10 μM and 25 μM) as a positive control. The leaves were soaked for another 3 h. The epidermis of the leaves was carefully torn off with tweezers and the mesophyll cells on the epidermis were washed with a brush. The clean epidermis was soaked in the above solutions, the size of the stomata was observed under an inverted fluorescence microscope in the bright field. The size of the stomata was measured with Image J software. Each replicate included approximately 30 stomatal counts and the experiment was repeated three times. 

Desiccation assays were carried out according to the method by Cao et al. [27]. Seven-day-old Col-0 wild-type plants with identical sizes were transferred to soil and maintained under short-day conditions (10 h light, 14 h dark) for another two weeks. Plants were sprayed with chemical solutions containing different concentrations of SLG1, using DMSO solution (0.02% Tween 20, 0.5% DMSO) and ABA (50 μM) as the control. After drying for 14–20 days, the plants were imaged, re-watered, and imaged after a day. 

Furthermore, to determine the rate of water loss from the leaves, two-week-old plants grown in the soil were sprayed with 50 μM of ABA and different concentrations of SLG1 and maintained for another 3 h in the light incubator. The rosette leaves of similar size were cut and weighed immediately. The detached leaves exposed to the air were weighed every 20 min. The experiment was repeated three times.

### 4.6. Gene Expression Analysis

*A. thaliana* seedlings for root length were used for gene expression analysis. Total RNA was extracted using the Plant RNA Kit (Beibei, Zhengzhou, China) following the manufacturer’s instructions, and 1 μg of RNA was used for cDNA synthesis using HiScript^®^ III RT SuperMix for qPCR (+gDNA wiper) (Vazyme, Nanjing, China). Finally, qRT-PCR was performed using SYBR Green (Vazyme, Nanjing, China) on a CFX96 Touch Real-Time PCR Detection System (Bio-rad, Berkeley, CA, USA) with the following hermocycling steps: 95 °C for 3 min, 45 cycles of 95 °C for 15 s, and 60 °C for 20 s. *A. thaliana actin2* was used as an internal control. The relative expression levels of genes were calculated using the 2^−ΔΔCt^ method with three biological replicates. The primers used were listed in Appendix A.

### 4.7. Stability Measurement of ABA and SLG1

ABA and SLG1 were dissolved in DMSO, diluted with water to 100 µM, and then filtered through a 0.45 µm filter to sterilize. Further, 200 µL of ABA and SLG1 were exposed to a UV light (8 W) [14]. Three biological replicates for each sample were performed. The contents of ABA and SLG1 were quantified on the Agilent 1290 InfinityII-6470 Triple four-stage rod liquid mass spectrometer [36]. 

### 4.8. Yeast Two-Hybrid Assay

Yeast two-hybrid assay was carried out as described previously [37]. Open reading frames (ORFs) of PYR1/PYL1–12 and three PP2Cs (ABI2, ABI1, and HAB1) were cloned using the cDNA of Col-0 plants and recombined into pGBKT7 vector and pGADT7 vector, respectively. The primers used for cloning are listed in Appendix A. The pGBKT7-PYR1/PYL1–12 vectors and the pGADT7-ABI2/ABI1/HAB1 vectors were co-transformed into the yeast strain AH109. After three days of incubation at 30 °C, the clone cultures were adjusted to an OD_600_ of 0.5 and then diluted five- and 25-folds. Then, 2 μL of the diluted cultures were dropped on SD agar plates lacking Leu, Trp (SD/−Leu/−Trp) to observe whether the plasmids were successfully transformed. Another 2 μL of the diluted cultures was dropped on SD agar plates lacking Leu, Trp, and His (SD/−Leu/−Trip/−His) containing 5 mM of 3-AT and 10 μM of ABA or different concentrations of SLG1 to incubate for another three days at 30 °C. Three clones of each combination were used in this assay and the experiment was performed three times.

### 4.9. Agonist/Receptor-Mediated Phosphatase Inhibition Assays

Furthermore, expression vectors were constructed to obtain the fusion proteins of PYR1/PYL1–12. Specifically, 8 out of the 13 ABA receptors in *A. thaliana* were expressed as 6x His-tagged in pET28a (PYR1, PYL1–6, and PYL8). PYL7 and PYL9–12 were expressed as MBP fusion proteins in the pMAL-c5x vector. ABI2 PP2C was cloned into the pGEX-4T-1 vector and expressed as GST fusion proteins. All vectors were transformed into the BL21 host cells. The successfully transformed cells were pre-cultured overnight, then transferred to a new LB medium at a ratio of 1:100 and cultured at 37 °C. At an OD_600_ of 0.6, 0.8 mM of isopropyl β-D-thiogalactoside (IPTG) was added to the culture and incubated for 4 h. Cells were harvested by centrifugation and lysed by sonication with Tris-HCl buffer. The recombinant His-tagged proteins were purified on Ni columns (QiAGEN, Hilden, GERMANY), the MBP-tagged fusion proteins were purified using amylose resin (New England Biolabs, Ipswich, MA, USA), and GST-tagged proteins were purified on GST resin (YEASEN, Shanghai, China). The PP2C activity assay of ABI2 was measured as described before ([18]). Then, 100 nM ABI2-GST and 200 nM PYR1/PYL1–12 fusion proteins were mixed with 10 μM ABA or 10 μM SLG1, placed in 80 μL buffer (100 mM Tris-Ace, pH 7.9, 100 mM NaCl, 3 μg BSA, and 0.1% β-mercaptoethanol), and then incubated at 30 °C for 30 min for a complete combination. Next, 20 μL of 4-methylumbelliferyl phosphate (20 mM) was added to react. The absorbance was measured immediately (λ_exc_ = 360 nm, λ_emm_ = 460 nm) on a Near Infrared Spectrometer. PP2C activity was calculated relative to solvent-only control wells (no test chemical). The experiment was repeated three times.

### 4.10. Molecular Docking

Crystal structures of PYL2 (PDB_ID: 3KDI) at a resolution of 2.379Å and PYL3 (PDB_ID: 4DSB) at a resolution of 2.70 Å were downloaded from the PDB database. These proteins were prepared according to the above-mentioned method with the active site size set to 10 Å. The structures of ABA and SLG1 were downloaded and prepared as mentioned above. Further, molecular docking was performed using the CDOCKER module of Discovery Studio software. The parameter settings of the CDOCKER module are as follows: ‘Pose Cluster Radius’ was set to 0.5 and other parameters are default. The optimal molecular conformation with the comprehensive score of ‘CDOCKER Energy’ and ‘CDOCKER Interaction Energy’ was selected to analyze the docking results.

### 4.11. Molecular Dynamics Simulations 

The above molecular docking results of ABA- or SLG1-PYLs complexes were saved in .pdb file format, respectively. The molecular dynamics (MD) simulations (50 ns) of all the complexes were done to access their stability and dynamics using GROMACS (GROningen Machine for Chemical Simulations, version 2021.2) [25,38]. The CHARMM36 force field was used to obtain the topology characteristics of PYLs. The CHARMM General Force Field (CGenFF) was used to obtain the topology characteristics and force field of the ligand. The ABA- or SLG1-PYLs complexes were solvated in a cubical box through the water model TIP3P. The overall systems were neutralized by adding the required number of Cl^−^ ions. Then, the energy of all system was minimized with the steepest descent of 50,000 steps. The NVT ensemble was used to balance the system temperature from 0 to 300 K at 100 ps, while the NPT ensemble was used to balance the system pressure to 1.0 bar at 100 ps. The LINCS (Linear Constraints Solver) algorithm was employed to restrain all bonds. Finally, 50 ns MD simulations were run and the trajectory was snapshotted/saved separately for each complex after every 2 fs. The complex’s root mean square deviation (RMSD), root mean square fluctuation (RMSF), the radius of gyration (Rg), and the solvent accessible surface area (SASA) were detected.

### 4.12. Statistical Analysis

In this study, all values are presented as the mean ± standard deviation (SD). One-way ANOVA and Tukey test were performed to analyze statistically significant differences between the treatments and the mock-treated controls (* *p* < 0.05, ** *p* < 0.01, *** *p* < 0.001, **** *p* < 0.0001) using SPSS software. Origin software was used for designing graphs. 

## Figures and Tables

**Figure 1 ijms-24-08783-f001:**
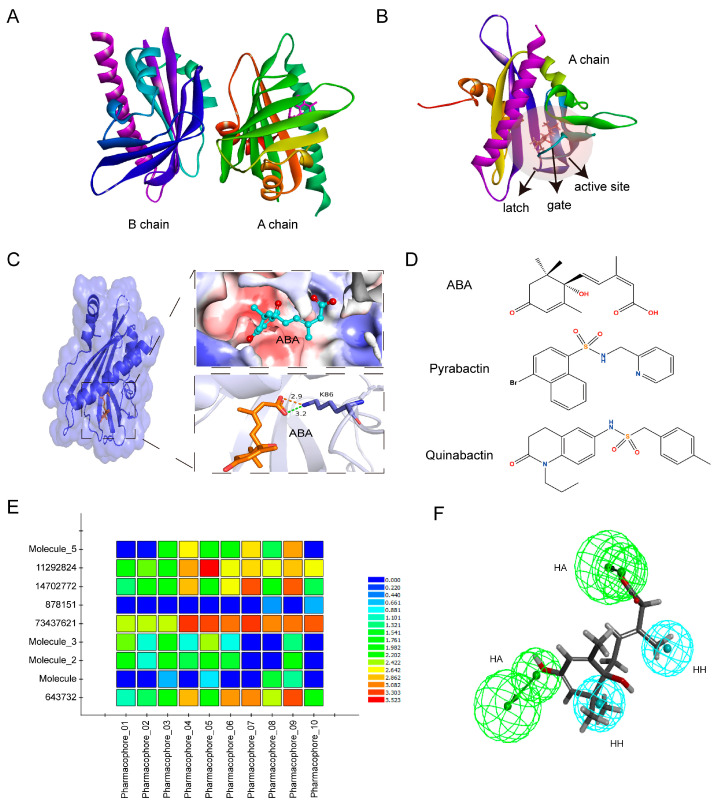
Virtual screening based on receptors and ligands. (**A**) Crystal structure of PYL1 protein with A and B chains (Protein Data Bank (PDB) ID: 3NEF). (**B**) The position of the active site in the A chain of PYL1. The red transparent ball represents the active site. (**C**) The three-dimensional (3D) structure of ABA-bond PYL1. ABA was located in a hydrophobic pocket, and its carboxylate formed a hydrogen bond and a charge interaction with a conserved lysine (K86) in PYL1 with a distance of 3.2 Å and 2.9 Å, respectively. The orange stick represents ABA. The upper right represents the hydrophobic pocket of the active site where ABA is located. Blue surface indicates hydrophilic residues and red surface indicates hydrophobic amino acid residues. The lower right represents the interaction between ABA and PYL1. The orange dashed line represents the charge interaction and the green dashed line represents the hydrogen bond interaction. The stick mode represents the small molecule ABA. The number represents the interaction distance in Å. (**D**) The molecular structure of ABA, pyrabactin, and quinabacin. (**E**) Matching of pharmacophore 0110 with test set compounds. The color indicates different scores, in which blue is the lowest and red is the highest. (**F**) Superposition of ABA and the optimal pharmacophore model 07. HA represents hydrogen bond acceptors in green balls and HH represents hydrophobic properties in blue balls.

**Figure 2 ijms-24-08783-f002:**
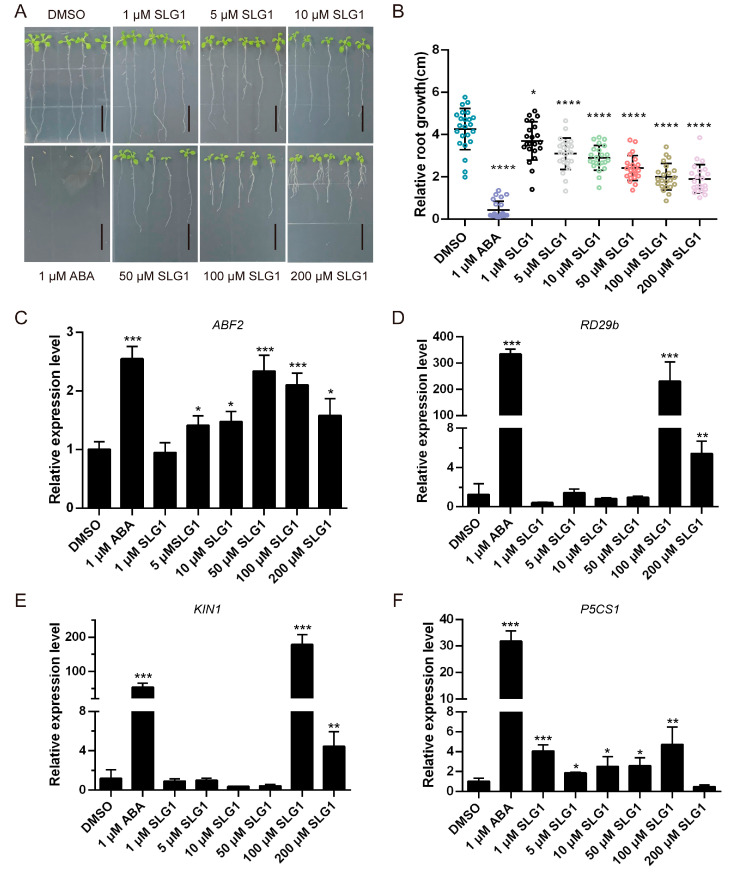
SLG1 inhibits the growth of *Arabidopsis thaliana* seedlings and activates the expression of ABA-related genes. (**A**) Seedlings of *A. thaliana* wild type (Col-0) were grown in 1/2 MS containing 1 μM of ABA or different concentrations of SLG1 for seven days, with 0.5% DMSO as control. The root length of seedlings in (**A**) was counted in (**B**). The column represents the mean value, and error bars indicate standard deviation (SD). Scale = 1 cm. (**C**–**F**) SLG1 potently activates ABA-related genes *ABF2, RD29b*, *KIN1*, and *P5CS1*. Three independent experiments were performed. One-way ANOVA and Tukey’s test were performed to determine the statistically significant difference (* *p* < 0.05, ** *p* < 0.01, *** *p* < 0.001, **** *p* < 0.0001).

**Figure 3 ijms-24-08783-f003:**
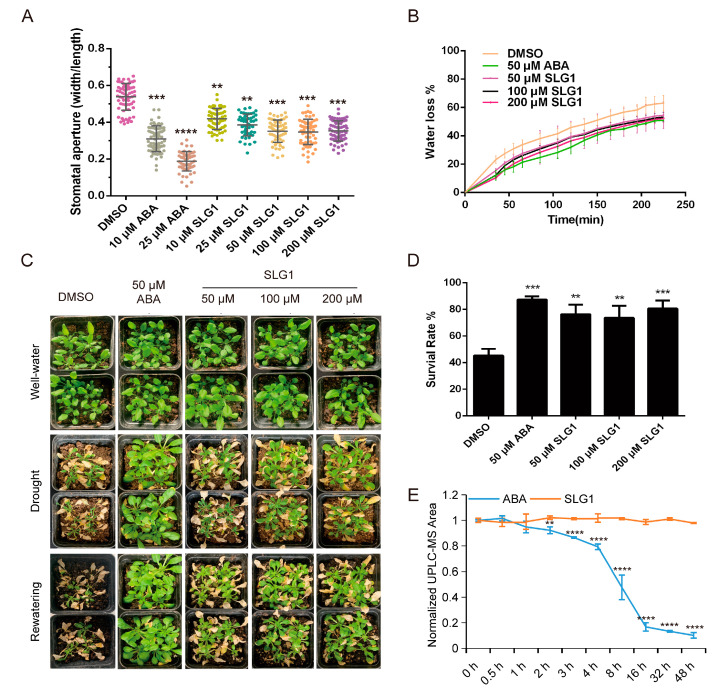
SLG1 treatment reduces stomatal aperture and water loss and enhances drought resistance. (**A**) Stomatal aperture assay. SLG1 was used at a concentration of 10 to 200 μM and (+)-ABA at 10 and 25 μM. Thirty stomata were analyzed in each experiment. (**B**) Water loss rate of rosette leaves separated from three-week-old plants. Well-irrigated plants were sprayed with SLG1 or (+)-ABA solutions, and rosette leaves of identical size were collected and analyzed at 3 h after treatment. Additionally, 0.5% DMSO was used as a control. The water loss rate was determined as % of initial fresh weight. (**C**) Drought stress assay. Wild-type (Col-0) plants grown under the short-day conditions for three weeks were used for drought stress assay, using 0.5% DMSO as a control. Images were captured before water withdrawal (**top** panel), before re-watering (**middle** panel), and one day after re-watering (**bottom** panel). (**D**) Survival rates in (**C**) were calculated. (**E**) Stable structure of SLG1 compared with ABA. Here, 100 µM of ABA and SLG1 were exposed to a UV light (8 W), and the content was measured at 0, 0.5, 1, 2, 3, 4, 8, 16, 32, and 48 h after exposure by LC-MS. The data represent the means ± SD of three biological repeats. One-way ANOVA and Tukey test were performed to analyze statistically significant differences between the treatments and the mock-treated controls (** *p* < 0.01, *** *p* < 0.001, **** *p* < 0.0001).

**Figure 4 ijms-24-08783-f004:**
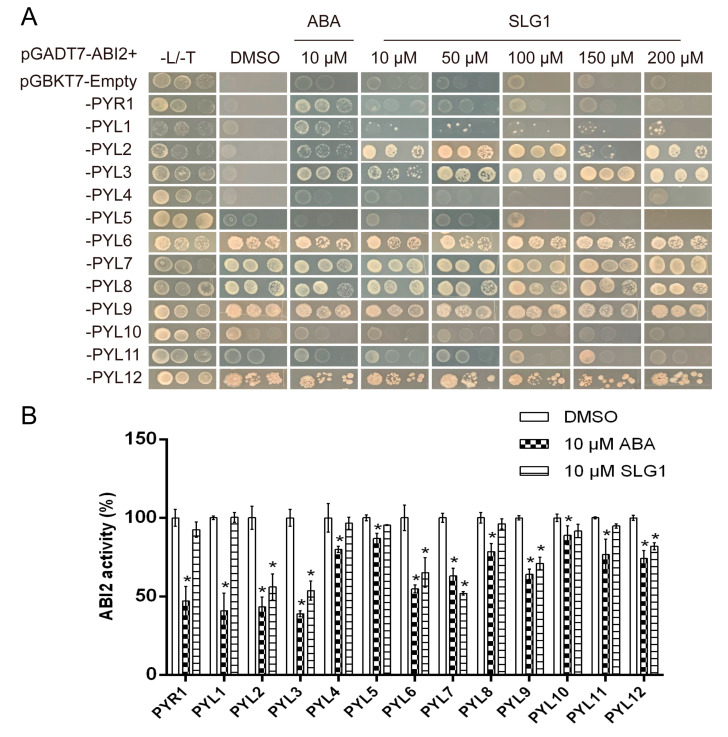
SLG1 is a potent agonist of ABA receptors. (**A**) SLG1 promotes the interaction of PYL2 and PYL3 with ABI2. Yeast two-hybrid assay showing the interactions of the binding domain (BD)-fused PYR1/PYLs with the activation domain (AD)-fused ABI2 on SD/−Leu/−Trp/−His (−L/−T/−H) media containing (+)-ABA and corresponding compounds. SD/−Leu/−Trp/−His media containing 0.1% DMSO was used as the negative control. Three replicates were maintained per concentration. (**B**) SLG1 induces the inhibition of ABI2 activity mediated by 13 PYLs, including PYR1 and PYL1–12, in phosphatase activity assay. The working concentration of SLG1 and (+)-ABA was 10 μM. Three replicates were maintained per concentration. One-way ANOVA and Tukey test were performed to analyze statistically significant differences between the treatments and the mock-treated controls (* *p* < 0.05).

**Figure 5 ijms-24-08783-f005:**
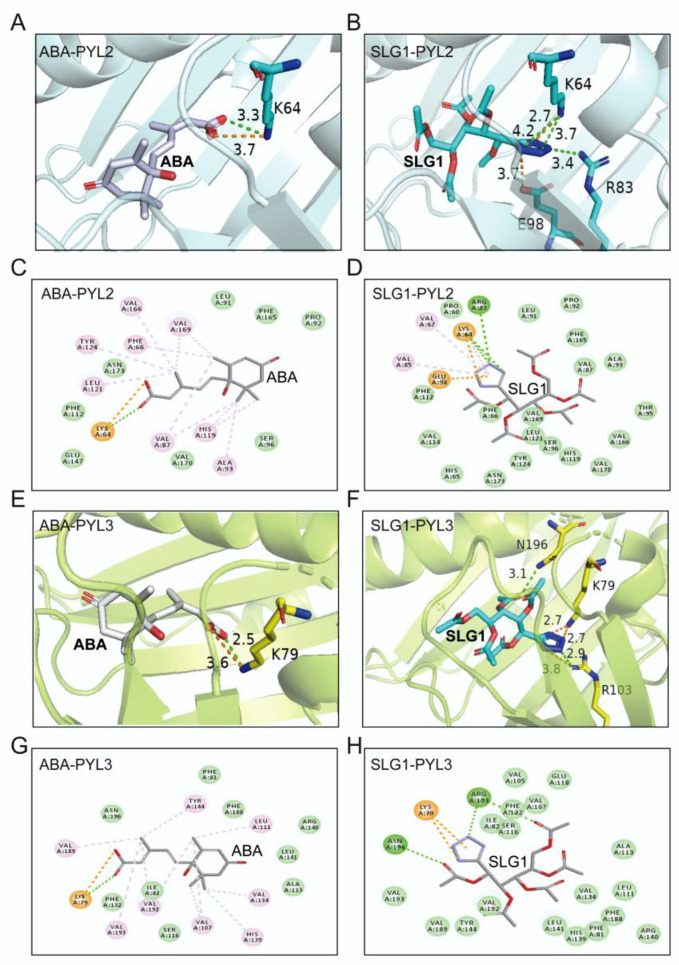
Molecular docking simulations of ABA and SLG1 with PYL2 and PYL3. (**A**,**B**) Three-dimensional diagram of the molecular docking between ABA/SLG1 and the key-residues of PYL2. (**C**,**D**) Two-dimensional diagram of the molecular docking between ABA/SLG1 and the key-residues of PYL2. (**E**,**F**) Three-dimensional diagram of the molecular docking between ABA/SLG1 and the key-residues of PYL3. (**G**,**H**) Two-dimensional diagram of the molecular docking between ABA/SLG1 and the key-residues of PYL3. The number represents the distance to form a key in Å. The green dotted lines represent conventional hydrogen bond interactions; orange dotted lines represent attractive charge interactions; light pink dotted lines represent alkyl interactions (hydrophobic interactions).

**Figure 6 ijms-24-08783-f006:**
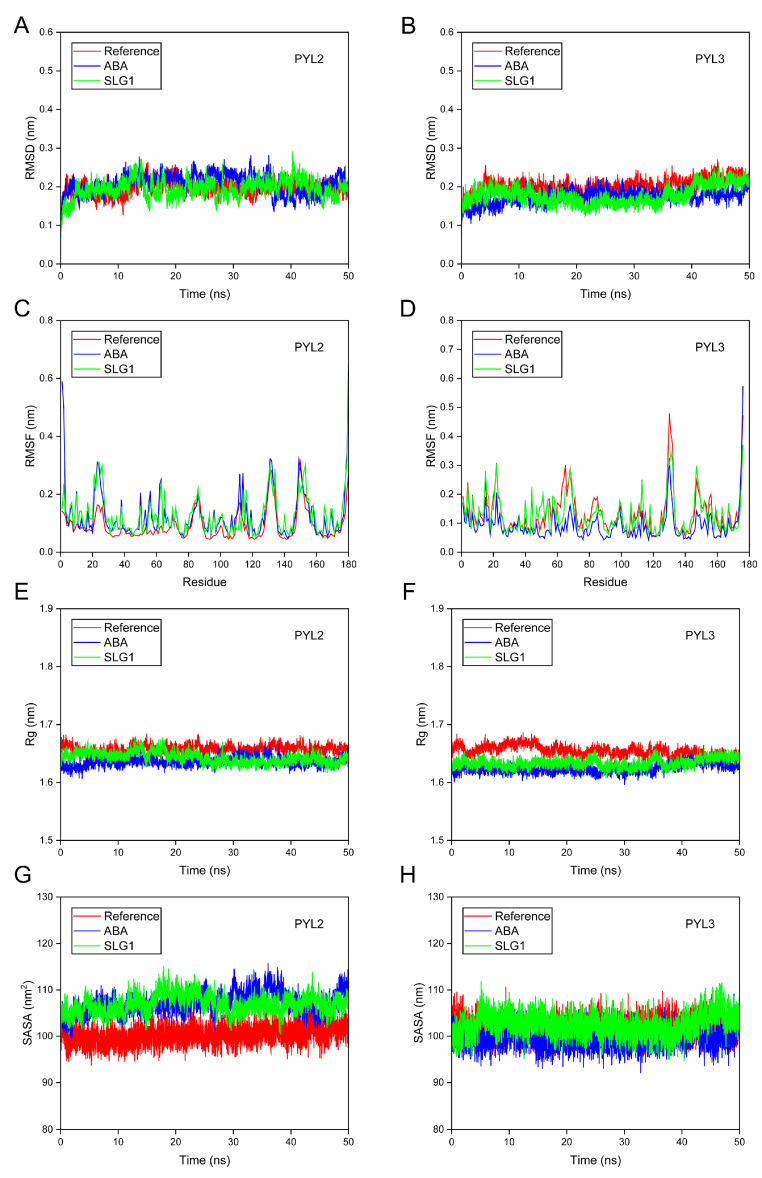
Molecular dynamics simulations of the interaction between ABA-/SLG1- PYL2/PYL3. (**A**,**B**) RMSD of PYL2 and PYL3 with ABA and SLG1. (**C**,**D**) RMSF of PYL2 and PYL3 with ABA and SLG1. (**E**,**F**) Rg of PYL2 and PYL3 with ABA and SLG1. (**G**,**H**) SASA of PYL2 and PYL3 with ABA and SLG1. The red line represents the reference (protein without ligand), the blue line represents ABA, and the green line represents SLG1.

**Figure 7 ijms-24-08783-f007:**
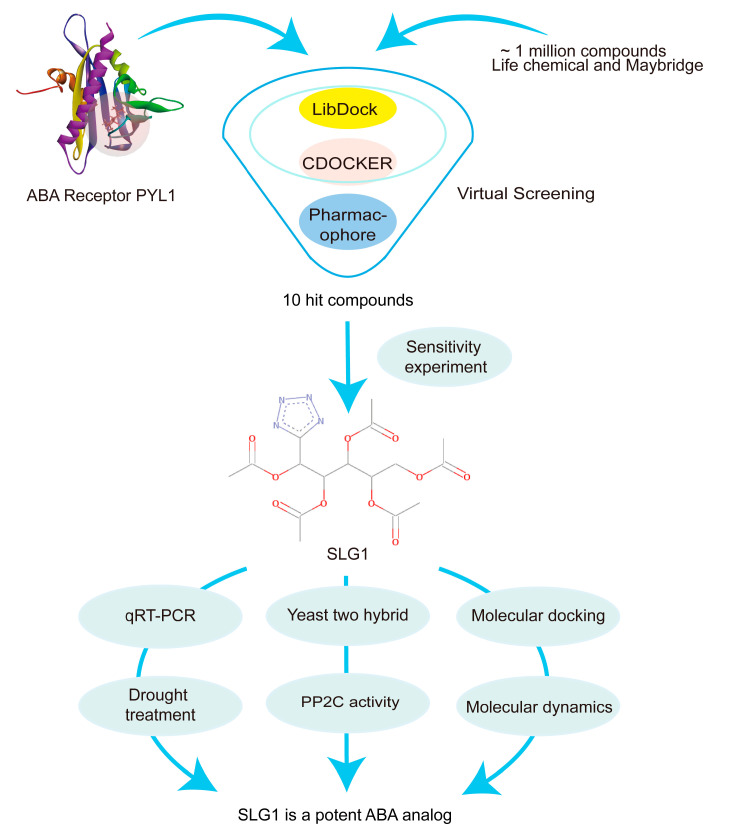
Flowchart of this study. Virtual screening methods, such as molecular docking and pharmacophore modeling, were used to screen small molecule compounds from a library of approximately 1,000,000 small molecules. Finally, an ABA analogue, SLG1, was identified with the ability to inhibit seedling growth, water loss, and desiccation.

## Data Availability

All data are contained within the article or Appendix A.

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
