# Peer review of "A Potential ABA Analog to Increase Drought Tolerance in Arabidopsis thaliana"

_ijms, 2023, doi:10.3390/ijms24108783_

Round 1
Reviewer 1 Report
I checked your manuscript and described comments below.
Abscisic acid (ABA) is an important phytohormone and is involved in dormancy, growth inhibition, and stomatal closure.
ABA analog SLG1 binds with PYL1 to inhibit the growth of seedlings and enhance drought resistance in A. thaliana.
I think you should fix the following points.
1. It would be easier to understand with the amino acid sequences and protein domain structures for PYL1-PYL12.
2. It would be better to include the following references for GROMACS.
Páll, et al. (2020) J. Chem. Phys. 153, 134110 (DOI:10.1063/5.0018516)
I don't think this paper has any major mistakes or grammatical problems.
Reviewer 2 Report
Dear editor and colleagues,
I have read with great interest the submitted manuscript “A potential ABA analog to increase drought tolerance in Arabidopsis thaliana”.
It is a study describing the in-silico analyses conducted for identifying ABA analogues and the application of the candidate moiety (SLG1) to Arabidopsis plants in order to test the hypothesis.
The work is exceptional in terms of experimentation and is pretty straightforward.
The authors have used appropriate techniques: in silico screening, modeling interaction & dynamics as well as in planta application.
The authors have proven that SLG1 has a better half time than ABA (although it is not as active in terms of concentration), and showed via a plethora of discrete methods (qPCR, root length, stomatal conductance, Yeast two-hybrid assay etc) the functional role of SLG1 in drought stress.
The manuscript per se is well organized, easy to follow and the conclusions are well supported by data produced; hence this work has merit for publication.
I have very few comments for the authors.
· There are some errors in terms of formatting (the authors should use italics for Latin words) L18, L133 etc; so, I suggest a proofreading before submission.
· Figure 1 should not be in the introduction but rather in materials and methods (M&M) section describing the screening scheme.
· I am curious if the authors removed the protein tags (for instance His-tag) in the experiments followed; because I did not observe that within text.
· There is not a section (in M&M) for the statistics used (for instance R & Anova?)
Based on the above, I recommend acceptance after minor revisions
Minor editing of English language required
Author Response
Dear Reviewers:
We are very grateful for your recognitions and comments concerning our
manuscript entitled “A potential ABA analog to increase drought tolerence in
Arabidopsis thaliana (ijms-2356361)”. You provided really valuable suggestions,
which are all very helpful for improving our manuscript. Based on your good
suggestions, we have made careful modifications. We hope that the revised
manuscript is more acceptable and satisfactory. Below you will find our
point-by-point responses to your comments:
Reviewer #2:
1. There are some errors in terms of formatting (the authors should use italics for
Latin words) L18, L133 etc; so, I suggest a proofreading before submission.
Response: Thanks very much for your suggestions. “A. thaliana” in the whole text
has been changed to “A. thaliana” in the clean manuscript.
2. Figure 1 should not be in the introduction but rather in materials and methods
(M&M) section .
Response: Thanks very much for your suggestions. Figure 1 describing the
screening scheme has been adjusted to new Figure 7 in materials and methods (M&M)
section in the revised manuscript.
3. I am curious if the authors removed the protein tags (for instance His-tag) in the
experiments followed; because I did not observe that within text.
Response: Thanks very much for your suggestions and meticulous revisions. We
expressed fusion proteins with His-tag (PYR1, PYL1–6, and PYL8), MBP-tag (PYL7
and PYL9–12) and GST-tag (ABI2) to measure ABI2 activity. In the revised
manuscript, the sentence has been changed to “100 nM ABI2-GST and 200 nM
PYR1/PYL1–12 fusion proteins were mixed with 10 μM ABA or 10 μM SLG1, placed in 80
μL buffer (100 mM Tris-Ace, pH 7.9, 100 mM NaCl, 3 μg BSA, and 0.1%
β-mercaptoethanol), and then incubated at 30°C for 30 min for a complete combination.”
4. There is not a section (in M&M) for the statistics used (for instance R & Anova?)
Response: Thanks very much for your suggestions and meticulous revisions. The
section (in M&M) for the statistics
“4.12. Statistical analysis
In this study, all values are presented as the mean ± standard deviation (SD). One-way
ANOVA and Tukey test were performed to analyze statistically significant differences
between the treatments and the mock-treated controls (* P < 0.05, ** P < 0.01, *** P
< 0.001, **** P < 0.0001)using SPSS software. Origin software was used for
designing graphs.” has been added in the revised manuscript.